# Phylogeny, Taxonomy and Morphological Characteristics of *Apiospora* (*Amphisphaeriales*, *Apiosporaceae*)

**DOI:** 10.3390/microorganisms12071372

**Published:** 2024-07-04

**Authors:** Congcong Ai, Zixu Dong, Jingxuan Yun, Zhaoxue Zhang, Jiwen Xia, Xiuguo Zhang

**Affiliations:** Shandong Provincial Key Laboratory for Biology of Vegetable Diseases and Insect Pests, College of Plant Protection, Shandong Agricultural University, Taian 271018, China; 15053892158@163.com (C.A.); 13173336553@163.com (Z.D.); 18765120619@163.com (J.Y.); zhangzhaoxue2022@126.com (Z.Z.); xiajiwen1@126.com (J.X.)

**Keywords:** *Apiospora*, taxonomy, *Amphisphaeriales*, phylogeny

## Abstract

*Apiospora* is widely distributed throughout the world, and usually identified as endophytes, pathogens or saprobes. In this study, six strains were isolated from *Bambusaceae* sp., *Prunus armeniaca*, *Salix babylonica* and saprophytic leaves in Shandong Province, China. Three new species were identified based on a multi-locus gene phylogenetic analysis using a combined dataset of ITS, LSU, TEF1α and TUB2 in conjunction with morphological assessments. *Apiospora armeniaca* sp. nov., *Apiospora babylonica* sp. nov., and *Apiospora jinanensis* sp. nov. have been comprehensively described and illustrated, representing significant additions to the existing taxonomy.

## 1. Introduction

*Apiospora* Sacc., which is a type genus of *Apiosporaceae* K.D. Hyde, J. Fröhl., Joanne E. Taylor & M.E. Barr, has been typified with *A. montagnei*, a new name for *Sphaeria apiospora* [1]. Most species of *Apiospora* are found in association with plants, either as endophytes, pathogens, or saprobes, with a wide host range and geographic distribution [2,3,4]. The distinguishing feature of the sexual morphs is their multi-locular perithecial stromata, adorned with hyaline ascospores that are enveloped in a robust gelatinous sheath [3,5]. The asexual form of *Apiospora* is noted for its basauxic conidiogenesis, featuring globose to subglobose conidia, typically appearing lenticular from the side, obovoid, and varying in color from pale brown to brown [3,6]. Additionally, certain species have been successfully isolated from a variety of sources including lichens, air, soil, and animal tissues [7,8,9,10,11]. Presently, there are 176 *Apiospora* records in the Index Fungorum (http://www.indexfungorum.org/, accessed on 1 June 2024).

In the last 5 years, *Apiosporaceae* has undergone multiple taxonomic revisions [5,12,13,14,15,16,17]. In a recent *Sordariomycetes* outline, Hyde et al. [12] identified five genera in the *Apiosporaceae* family: *Appendicospora*, *Arthrinium*, *Dictyoarthrinium*, *Endocalyx*, and *Nigrospora*. Soon after, *Dictyoarthrinium* was moved to *Didymosphaeriaceae*, following a multi-locus gene phylogenetic analysis [13]. Pintos and Alvarado [5] later distinguished *Apiospora* from *Arthrinium*, after studying the type species of both genera and conducting multigene phylogeny. Recently, Konta et al. [14] reclassified *Endocalyx* into *Cainiaceae*, based on both morphological and phylogenetic evidence. Additionally, Samarakoon et al. [15] introduced the novel family *Appendicosporaceae* specifically for *Appendicospora*. Furthermore, Wijayawardene et al. [18] acknowledged five genera within this family—*Appendicospora*, *Apiospora*, *Arthrinium*, *Dictyoarthrinium*, and *Nigrospora*. *Apiospora* and *Arthrinium* 1 are considered synonymous, with the former denoting the sexual form and the latter denoting the asexual form in dual nomenclature [19]. Following the shift from dual nomenclature, the previous name *Arthrinium* has been suggested for incorporation into singular nomenclature [19]. The *Arthrinium* genus was first introduced by Kunze and Schmidt in 1817 and later sanctioned by Fries in 1832, having *Arthrinium caricicola* assigned as the type species [20,21]. Recent studies involving various genes have shown that *Arthrinium* and *Apiospora* constitute two distinct, clearly connected lineages closely related to *Nigrospora* within *Apiosporaceae* [5,15].

Some species of *Apiospora* are important plant pathogens; for example, *Apiospora arundinis* is a fungus that causes brown culm streak in bamboo, chestnut leaf spot, and barley kernel blight [22,23,24], with *Apiospora sacchari* causing damping off in durum wheat [25]. The aim of this study was to clarify the taxonomic status of *Apiospora* and conduct morphological and phylogenetic studies. We described three new species, viz, *Apiospora armeniaca* sp. nov., *Apiospora babylonica* sp. nov. and *Apiospora jinanensis* sp. nov., based on molecular phylogenetic analyses and morphological observations.

## 2. Materials and Methods

### 2.1. Isolation and Morphological Study

During a series of field visits in 2022 in Shandong province in China, specimens displaying necrotic spots were collected, and single spore isolation and tissue isolation techniques were used to obtain a single colony [26]. Culture tissue fragments were grown on Potato Dextrose Agar (PDA: 220 g potato, 20 g agar, 18 g dextrose, 1000 mL sterile water, and natural pH) at 25 °C for 3 days. Morphologically distinct colonies appeared on the PDA plate. The mycelium showing vigorous growth at the colony’s edge was isolated and transferred to a fresh PDA plate for further cultivation at 25 °C. Images were captured using a Sony Alpha 6400L digital camera (Sony Group Corporation, Tokyo, Japan) on days 7 and 14. Microscopic examination of the fungal structures was conducted using an Olympus SZ61 stereo microscope and an Olympus BX43 microscope (Olympus Corporation, Tokyo, Japan), along with a BioHD-A20c color digital camera (FluoCa Scientific, Shanghai, China) for recording.

All fungal strains were preserved in 15% sterilized glycerol at 4 °C, with each strain stored in three 2.0 mL tubes for future studies. Structural measurements were carried out using Digimizer software (v5.6.0) (this software comes with a statistical module that can calculate the mean and SD), with a minimum of 25 measurements taken for each characteristic, including conidiophores, conidiogenous cells, and conidia. Specimens were submitted to the Herbarium of Plant Pathology, Shandong Agricultural University (HSAUP) and Herbarium Mycologicum Academiae Sinicae (HMAS), with living cultures stored in the Shandong Agricultural University Culture Collection (SAUCC) for preservation and further research. The taxonomic details of the new species were provided to MycoBank (http://www.mycobank.org, accessed on 1 June 2024). All taxonomic data regarding the new species described in this study have been shared with MycoBank (http://www.mycobank.org, accessed on 1 June 2024).

### 2.2. DNA Extraction and Amplification

Fungal DNA was extracted from fresh mycelia grown on PDA using either the CTAB method or a kit method (OGPLF-400, GeneOnBio Corporation, Changchun, China) [27,28]. Four molecular markers, namely the ITS, LSU, TEF1α, and TUB2 genes, were amplified with specific primer pairs, as detailed in Table 1 [29,30,31,32]. The polymerase chain reaction was carried out using the Eppendorf Master Thermocycler from Hamburg, Germany. The amplification process involved a 20 μL reaction mix containing 10 μL 2 × Hieff Canace^®^ Plus PCR Master Mix (With Dye) from Yeasen Biotechnology (Shanghai, China) (Cat No. 10154ES03), 0.8 μL of each forward and reverse primer (TsingKe, Qingdao, China) at a concentration of 10 μM, and 1.0 μL of the genomic DNA template. The volume was adjusted to 20 μL with distilled deionized water. The PCR products were separated using 1% agarose gel electrophoresis with GelRed and visualized under UV light. Subsequently, gel recovery was performed using a Gel Extraction Kit (Cat: AE0101-C) from Shandong Sparkjade Biotechnology Co., Ltd. (Zouping, China) The purified PCR products underwent bidirectional sequencing by Biosune Company Limited (Shanghai, China). The raw data (trace data) were analyzed with MEGA v. 7.0 to ensure consistent sequences [33]. All sequences generated during this study were deposited in GenBank under the accession numbers provided in Appendix A.

### 2.3. Phylogenetic Analyses

The generated consensus sequences were subjected to Megablast searches to identify closely related sequences in the NCBI’s GenBank nucleotide database. New sequences generated in this study were aligned with pertinent sequences obtained from GenBank (Table 1), utilizing the MAFFT 7 online services and the default approach (http://mafft.cbrc.jp/alignment/server/, accessed on 1 June 2024) for comparison with other associated sequences. For the species-level identification of isolates, a separate phylogenetic analysis of each marker was first conducted and then combined (ITS-LSU-TEF1α-TUB2) (Refer to Appendix A). The multi-labeled data was analyzed phylogenetically using Bayesian inference (BI) and maximum likelihood (ML) algorithms. Both ML and BI analyses were performed on the CIPRES Science Gateway portal (https://www.phylo.org/, accessed on 1 June 2024) or offline software (The ML was operated in RaxML-HPC2 on XSEDE v8.2.12 and BI analysis was operated in MrBayes v3.2.7a with 64 threads on Linux) [30,31,32,33,34,35]. For ML analyses, we used default parameters and conducted 1000 rapid bootstrap replicates with the GTR+G+I model of nucleotide evolution. The BI analysis was carried out using a quick bootstrap algorithm with an automatic stop option. It comprised 2 million generations across sixty-four parallel runs with stop rule options and a 100-generation sampling frequency. The burn-in score was designated at 25%, and posterior probabilities (PP) were computed from the remaining trees. The visual representation of the resulting trees was performed using FigTree v. 1.4.4 or ITOL: Interactive Tree of Life. The tree designs were crafted in Adobe Illustrator CS6.

## 3. Results

### 3.1. Phylogenetic Analyses

Phylogenetic analysis was performed on 107 isolates of *Apiospora* species. Of these, 106 isolates were categorized as the ingroup, while one strain of *Arthrinium caricicola* (CBS 145127) was defined as the outgroup. The total alignment included 2843 concatenated characters, distributed across ranges 1–831 (ITS), 832–1673 (LSU), 1674–2244 (TEF1α), and 2245–2792 (TUB2). Among these characters, 1388 remained constant, 511 were variable and parsimony-uninformative, and 893 were parsimony-informative. There existed 1705 distinct alignment patterns, with gaps and wholly undetermined characters accounting for 31.31% of the alignment. The final ML Optimization Likelihood was −28,478.957873. The estimated base frequencies were as follows: A = 0.233901, C = 0.250499, G = 0.258001, T = 0.257599; substitution rates AC = 1.359081, AG = 2.638423, AT = 1.095604, CG = 0.952105, CT = 4.645088, and GT = 1.000000; gamma distribution shape parameter α = 0.337980. The GTR+I+G model was proposed for ITS, LSU, TEF1α, and TUB2. MCMC analysis of these four tandem genes was performed over 16,990,000 generations across 254,852 trees. The initial 84,950 trees, representing the burn-in phase of the analysis, were excluded, with the remaining trees used for calculating the posterior probability in the majority rule consensus tree (Figure 1; first value: PP > 0.90 shown). The alignment comprised a total of 1711 unique site patterns (ITS: 469, LSU: 486, TEF1α: 409, TUB2: 347). The topology of the ML tree confirms the tree topology obtained from Bayesian inference; therefore, only the ML tree is presented (Figure 1).

The 107 strains were classified into 95 species based on the phylogeny of five genes (Figure 1). In our phylogenetic analyses, 106 strains of Apiospora formed a monophyletic clade (Figure 1). Within them, six strains represented three new species lineages: *Apiospora armeniaca* (SAUCC DL1831, SAUCC DL1844), closely related to *A. guizhouensis* (LC5322) with full support (98% MLBV and 1.0 BIPP); *A. babylonica* (SAUCC DL1841, SAUCC DL1864), closely related to *A. bawanglingensis* (SAUCC BW0444), *A. indocalami* (SAUCC BW0455) and *A. piptatheri* (CBS 145149) with good support (74% MLBV and1.0 BIPP); and *A. jinanensis* (SAUCC DL1981, SAUCC DL2000), closely related to *A. italica* (CBS 145138) with good support (81% MLBV and1.0 BIPP). The present study revealed three species, viz. *Apiospora armeniaca* sp. nov., *A. babylonica* sp. nov., and *A. jinanensis* sp. nov.

### 3.2. Taxonomy

#### 3.2.1. *Apiospora armeniaca* H. Sheng, Z.X. Zhang & X.G. Zhang, sp. nov.

MycoBank—No: 854155

Etymology—Referring to the species name of the host plant *Prunus armeniaca*.

Type—China, Shandong Province, Kunyu Mountain National Nature Reserve, on leaves of *Prunus armeniaca*, 15 June 2022, Z.X. Zhang (HMAS 352687, holotype), ex-holotype living culture SAUCC DL1831.

Description—On PDA, hyphae 1.4–2.9 μm in diameter, hyaline, branched, and septate. Asexual morphology: Conidiophores are cylindrical, septate, verrucose, flexuous, and sometimes reduced to conidiogenous cells. Conidiogenous cells are indistinct, clustered on hyphae, and hyaline are pale brown, measuring 5.3–6.8 × 4.1–5.6. Conidia are brown to dark brown, smooth to finely roughened, and ranging from globose to subglobose to lenticular. They exhibit a longitudinal germ slit and occasionally elongate to ellipsoidal, 6.2–7.3 × 4.8–6.1 μm, mean ± SD = 6.7 ± 0.5 × 5.5 ± 0.6 μm, L/W = 1.1–1.2, n = 40. Sexual morph: Undetermined. See Figure 2.

Culture characteristics—Cultures incubated on PDA at 25 °C in darkness, reaching 81.7–83.5 mm diam., with a growth rate of 11.6–11.9 mm/day after 7 days; the colonies on PDA have regular edges, abundant white to gray aerial hyphae, flocculent cotton, and upright clusters of hyphae.

Additional specimen examined—China, Shandong Province, Kunyu Mountain National Nature Reserve, on leaves of *Prunus armeniaca*, 15 June 2022, Z.X. Zhang, HSAUP DL1844, living culture SAUCC DL1844.

Notes—Phylogenetic analyses of four combined genes (ITS, LSU, TEF1α, and TUB2) showed that *Apiospora armeniaca* sp. nov. formed an independent clade that is closely related to *A. guizhouensis* (LC5322) and *A. sacchari* (CBS 212.30). The *A. armeniaca* is distinguished from *A. guizhouensis* by 6/600, 2/813, 7/426 and 2/751 characters, from *A. sacchari* by 7/589, 3/831, 29/427 and 21/442 characters, and from *A. cordylines* by 6/600, 2/813, 25/441 and 8/763 characters in the ITS, LSU, TEF1α and TUB2 sequences, respectively. Morphologically, *A. armeniaca* differs from *A. guizhouensis* and *A. sacchari* in conidia (6.2–7.3 × 4.8–6.1 vs. 5.0–7.5 × 4.0–7.0 vs. 6–8 × 3.5–4 μm) [5,34]. Therefore, we identify it as a novel species.

#### 3.2.2. *Apiospora babylonica* H. Sheng, Z.X. Zhang & X.G. Zhang, sp. nov.

MycoBank—No: 854156

Etymology—Referring to the species name of the host plant *Salix babylonica*.

Type—China, Shandong Province, Dongying Botanical Garden, on diseased leaves of *Salix babylonica*, 17 July 2022, Z.X. Zhang (HMAS 352688, holotype), ex-holotype living culture SAUCC DL1841.

Description—On PDA, hyphae 2.2–4.0 μm diam, hyaline, branched, septate. Asexual morphology: Conidiophores are cylindrical, septate, verrucose, flexuous, and sometimes reduced to conidiogenous cells. The conidiogenous cells are indistinct, clustered on hyphae and hyaline, measuring 7.1–8.2 × 6.2–6.8 μm. The conidia are brown to dark brown, smooth to finely roughened, globose, subglobose to lenticular, have a longitudinal germ slit, and are occasionally elongated to ellipsoidal, 6.7–8.5 × 4.5–7.7 μm, mean ± SD = 7.6 ± 0.6 × 5.9 ± 0.9 μm, L/W = 1.2–1.4, n = 40. Sexual morph: Undetermined. See Figure 3.

Culture characteristics—Cultures incubated on PDA at 25 °C in darkness, reaching 82.8–84.7 mm diam., with a growth rate of 11.8–12.1 mm/ day after 7 days; the colonies on PDA plates have uniform white irregular edges, aerial white to gray hyphae, and a flocculent cotton-like texture.

Additional specimen examined—China, Shandong Province, Dongying Botanical Garden, on saprophytic leaves, 17 July 2022, Z.X. Zhang, HSAUP DL1864, living culture SAUCC DL1864.

Notes—Phylogenetic analyses of four combined genes (ITS, LSU, TEF1α, and TUB2) showed that *Apiospora babylonica* sp. nov. formed an independent clade that is closely related to *A. bawanglingensis* (SAUCC BW0444), *A. indocalami* (SAUCC BW0455) and *A. piptatheri* (CBS 145149). The *A. babylonica* is distinguished from *A. bawanglingensis* by 8/612, 2/817, 5/450 and 2/435 characters, from *A. indocalami* by 4/634, 2/819, 3/444 and 7/772 characters and from *A. piptatheri* by 10/579, 7/816, 50/448 and 0/435 characters in the ITS, LSU, TEF1α and TUB2 sequences, respectively. Morphologically, *A. bawanglingensis* differs from *A. bawanglingensis*, *A. indocalami* and *A. piptatheri* in conidia (6.7–8.5 × 4.5–7.7 vs. 6–8 × 3–5 vs. 7.3–8.9 × 5.7–8.6 vs. 6.8–8.9 × 5.7–7.8μm) [35]. *A. babylonica* conidia are smooth to finely roughened, while *A. indocalami* conidia are smooth; *A. babylonica* conidia are brown to dark brown, while *A.bawanglingensis* conidiogenous cells are dark green, becoming brown; *A. babylonica* conidiogenous cells are indistinct, aggregated in clusters on hyphae and hyaline, while the conidiogenous cells of *A. piptatheri* are discrete and sometimes branched [35]. Therefore, we identify it as a novel species.

#### 3.2.3. *Apiospora jinanensis* H. Sheng, Z.X. Zhang & X.G. Zhang, sp. nov.

MycoBank—No: 854157

Etymology—Referring to the location of the holotype, Jinan Botanical Garden.

Type—China, Shandong Province, Jinan Botanical Garden, on diseased leaves of *Bambusaceae* sp., 13 October 2022, Z.X. Zhang (HMAS 352689, holotype), ex-holotype living culture SAUCC DL1981.

Description—On PDA, hyphae 2.9–4.6 μm diam, hyaline, branched, septate. Asexual morph: Conidiophores, septate, flexuous, sometimes reduced to conidiogenous cells. Conidiogenous cells indistinct, aggregated in clusters on hyphae and hyaline, at 5.6–7.9 × 4.2–6.6 μm. Conidia are brown to dark brown, smooth to finely roughened, globose, subglobose to lenticular, possess a longitudinal germ slit, are occasionally elongated to ellipsoidal, and measure 5.7–6.9 × 5.2–6.7 μm, with a mean ± SD of 6.3 ± 0.3 × 5.6 ± 0.3 μm, L/W ratio of 1.1–1.2, based on a sample size of 40. Sexual morph: Undetermined. See Figure 4.

Culture characteristics—Cultures incubated on PDA at 25 °C in darkness, occupying an entire 90 mm petri dish after 7 days; the colonies on PDA have regular edges, and the aerial hyphae are white to gray, clustered into clusters, with a flocculent cotton-like texture; from the surface and back center to the edge of the colony, it is white to gray.

Additional specimen examined—China, Shandong Province, Jinan Botanical Garden, on diseased leaves of *Bambusaceae* sp., 13 October 2022, Z.X. Zhang, HSAUP DL2000, living culture SAUCC DL2000.

Notes—Phylogenetic analyses of four combined genes (ITS, LSU, TEF1α, and TUB2) showed that *Apiospora jinanensis* sp. nov. formed an independent clade that is closely related to *A. italica* (CBS 145138). The *A. jinanensis* is distinguished from *A. italica* by 4/575, 5/821, 12/431 and 23/741 characters in the ITS, LSU, TEF1α and TUB2 sequences, respectively. Morphologically, *A. jinanensis* is bigger than *A. italica* in conidia (5.7–6.9 × 5.2–6.7 vs. 4–6 × 3–4 μm) [5]. Therefore, we identify it as a novel species.

## 4. Discussion

The establishment of the family *Apiosporaceae* by Hyde et al. in 1988 to encompass the clade consisting of *Appendicospora*, *Apiospora*, *Arthrinium*, *Dictyoarthrinium* and *Nigrospora* within the *Amphisphaeriales* highlights the importance of phylogenetic analysis in understanding the evolutionary relationships among fungi [6,18]. The classification of Apiosporaceae has been revised [5,12,13,14,15,16,17]. Three new species are introduced and described in this study: *Apiospora armeniaca* sp. nov., *Apiospora babylonica* sp. nov., and *Apiospora jinanensis* sp. nov. The descriptions are based on their morphological characteristics and phylogenetic status. *Apiospora armeniaca* was isolated from two specimens (DL1831 and DL1844) and the leaf veins of *Prunus armeniaca* in Kunyu Mountain National Nature Reserve, Shandong Province, China, and two specimens displayed no apparent lesions. *Apiospora babylonica* was isolated from specimens of *Salix babylonica* and saprophytic leaves in Dongying Botanical Garden, Shandong Province, China, and we isolated it from diseased and saprophytic leaves also. *Apiospora jinanensis* was isolated from two specimens of *Bambusaceae* sp. in Jinan Botanical Garden, Shandong Province, China; in fact, most species of *Apiospora* were isolated from *Bambusaceae* sp. [36].

Currently, the Global Biodiversity Information Facility (GBIF, https://www.gbif.org/, accessed on 1 June 2024) holds 943 georeferenced records of *Apiospora* species worldwide. *Apiospora* thrives in subtropical, tropical, temperate, and cold regions spanning Africa, America, Asia, Australia, and Europe [37,38]. As an endophyte, plant pathogen, and saprophytic fungus, *Apiospora* thrives in various terrestrial environments, including soil, the atmosphere, and marine substrates. Its primary hosts are plants, especially *Poaceae* [39]. Only 16 *Apiospora* records have been isolated from woody plants (trees, shrubs, small shrubs), representing less than 10% of all records. Approximately half of these hosts belong to the *Arecaceae* family, based on existing statistical data from the USDA fungal database (https://nt.arsgrin.gov/fungaldatabases/, accessed on 1 June 2024) and a compilation of related literature published later on regarding the genus *Apiospora* [35]. In this study, *Apiospora armeniaca* was identified as an endophyte, as it was isolated from healthy leaves (*Prunus armeniaca*). *Apiospora babylonica* and *Apiospora jinanensis* were isolated from diseased leaves, but their pathogenicity needs to be verified. Therefore, we refer to it as an endophytic fungus associated with lesions.

*Apiospora* was known to thrive in a variety of habitats, from forest floors to grasslands, and was found on decaying organic matter. Its ability to adapt to different environmental conditions makes it a versatile and resilient fungus. We will continue to study *Apiospora* to better understand its role in ecosystems and its potential applications in biotechnology and medicine.

## Figures and Tables

**Figure 1 microorganisms-12-01372-f001:**
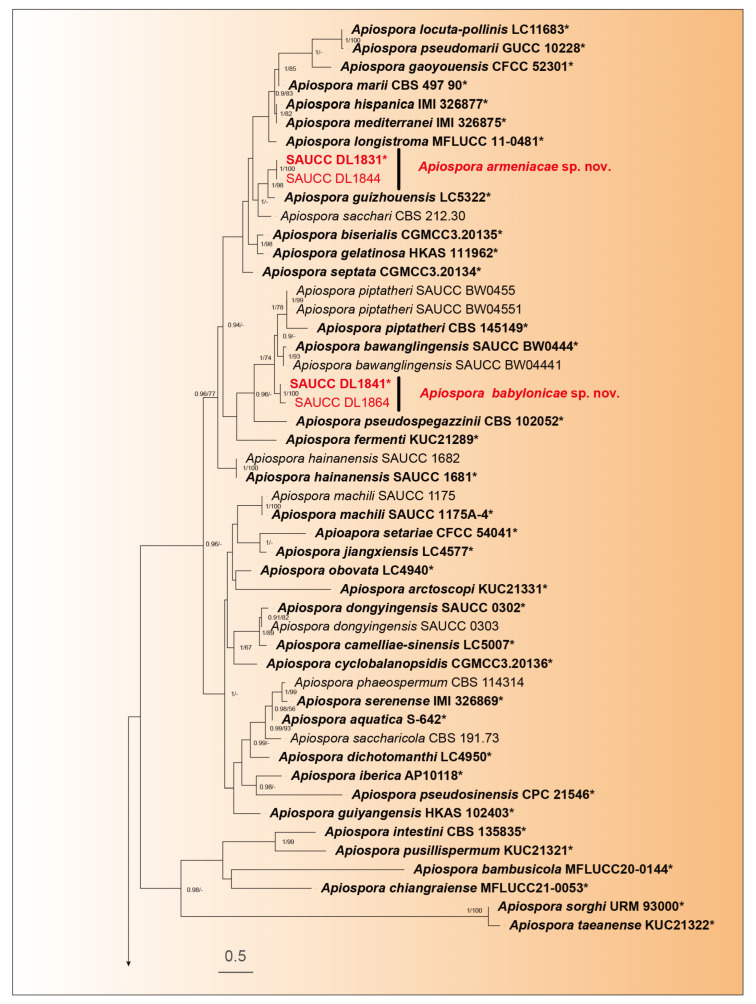
A maximum likelihood tree was constructed using a combined dataset of the analyzed ITS, LSU, TEF1α, and TUB2 sequences, and the roots on the *Arthrinium caricicola* (CBS 145127). The branch support values, shown as ML/BIPP, are indicated above the nodes: BIPP ≥ 0.90 on the left and MLBV ≥ 70% on the right are shown as BIPP/ML above the nodes. Strains marked with a star “*”and bolded represented are ex-types or ex-holotypes. Ex-type cultures are indicated in bold face while strains from the present study are shown in red. The scale at the bottom middle represents 0.5 substitutions per site. To enhance the visual appeal of the evolutionary tree layout, certain branches are shortened by two diagonal lines („/”), as indicated.

**Figure 2 microorganisms-12-01372-f002:**
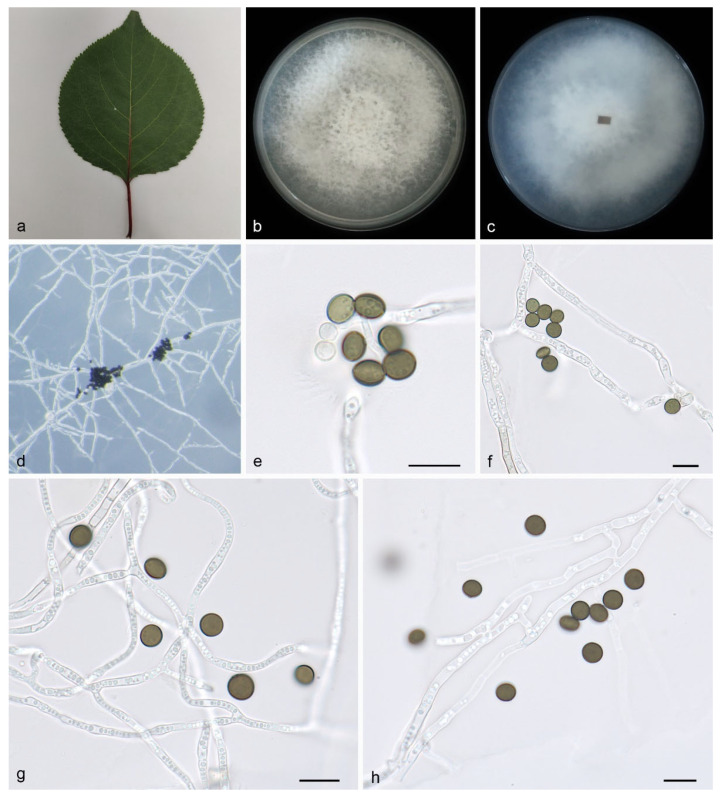
*Apiospora armeniaca* (HMAS 352687, holotype): (**a**) leaves of host plant; (**b**,**c**) colonies on PDA from above and below after 14 days; (**d**) colony overview; (**e**–**h**) conidia with conidiogenous cells. Scale bars: 10 μm. (**e**–**h**).

**Figure 3 microorganisms-12-01372-f003:**
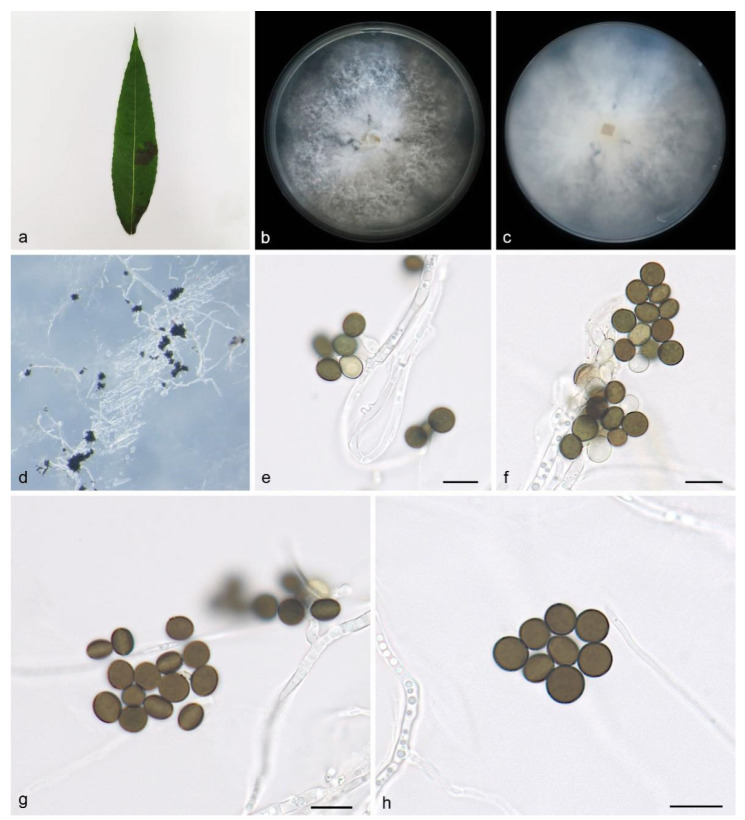
*Apiospora babylonica* (HMAS 352688, holotype): (**a**) leaves of host plant; (**b**,**c**) colonies on PDA from above and below after 14 days; (**d**) colony overview; (**e**,**f**) conidia with conidiogenous cells; (**g**,**h**) conidia. Scale bars: 10 μm. (**e**–**h**).

**Figure 4 microorganisms-12-01372-f004:**
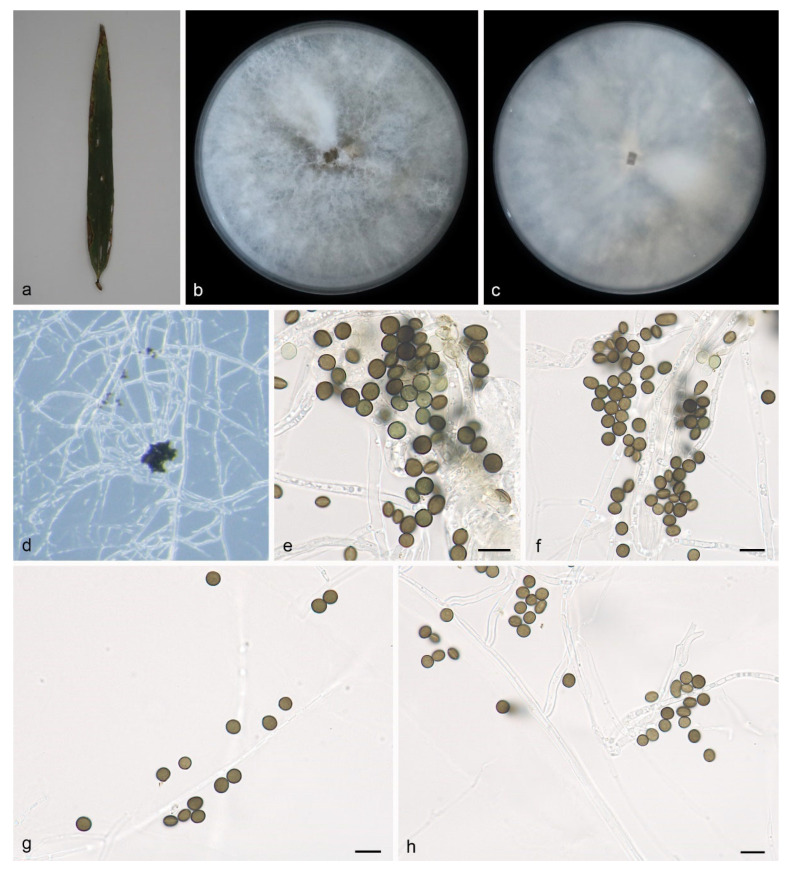
*Apiospora jinanensis* (HMAS 352689, holotype): (**a**) leaves of host plant; (**b**,**c**) colonies on PDA from above and below after 14 days; (**d**) colony overview; (**e**,**f**) conidia with conidiogenous cells; (**g**,**h**) conidia. Scale bars: 10 μm. (**e**–**h**).

**Table 1 microorganisms-12-01372-t001:** Molecular markers, PCR primers, and programs utilized in this investigation.

Loci	PCR Primers	Sequence (5′–3′)	PCR Cycles	Reference
ITS	ITS5	GGA AGT AAA AGT CGT AAC AAG G	(95 °C: 30 s, 55 °C: 30 s, 72 °C: 45 s) × 30 cycles	[29]
	ITS4	TCC TCC GCT TAT TGA TAT GC		
LSU	LR0R	GTA CCC GCT GAA CTT AAG C	(95 °C: 30 s, 48 °C: 50 s, 72 °C: 1 min 30 s) × 35 cycles	[30]
	LR5	TCC TGA GGG AAA CTT CG		
TEF1α	EF1-728F	CAT CGA GAA GTT CGA GAA GG	(95 °C: 30 s, 51 °C: 30 s, 72 °C: 1 min) × 35 cycles	[31]
	EF2	GGA RGT ACC AGT SAT CAT GTT		
TUB2	T1	AAC ATG CGT GAG ATT GTA AGT	(95 °C: 30 s, 56 °C: 30 s, 72 °C: 1 min) × 35 cycles	[32]
	Bt-2b	ACC CTC AGT GTA GTG ACC CTT GGC		

## Data Availability

The sequences from the present study were submitted to the NCBI database (https://www.ncbi.nlm.nih.gov/, accessed on 1 June 2024) and the accession numbers were listed in Appendix A.

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
