# Peer review of "Phylogeny, Taxonomy and Morphological Characteristics of Apiospora (Amphisphaeriales, Apiosporaceae)"

_microorganisms, 2024, doi:10.3390/microorganisms12071372_

Round 1
Reviewer 1 Report
Comments and Suggestions for Authors
To summarize the manuscript, three species are described from leaves of three plants in China. No original approach was used, isolates were presumably obtained from leaves on the tree or in the litter. Authors amplified three gene regions and reconstructed phylogeny of Apiospora, where their isolates formed three distinct lineages. Species descriptions are accompanied with photo documentation and description of morphology.
Unfortunately, I cannot recommend the manuscript to publication, most probably not even to resubmission due to serious mishandling with data. Below is a list of more or less serious conflicts that are apparent from the manuscript or from a simple BLAST search that I made using the published sequences.
Primarily, it is not know, what were the “Specimens” collected by authors (l. 61). Most probably leaves with necrotic spots, but since sentence on l. 62 is incomplete, it is not clear, if the cultures were obtained by Apiospora conidia picked from the necroses, or from cultured tissue fragments. What was then deposited as type specimens? Remaining necrotic leaves after isolation? Further in the text, there is no mention about “necroses”, but “diseased” leaves and conflicting information are provided for “A. armeniaca”, where it is stated that it originated from “diseased leaves” (l. 172), whilst on l. 282 and 299 it is explicitly stated that they were “isolated from healthy leaves”.
Four DNA regions were sequenced, two of them represent coding genes. TEF has however different name and abbreviation that should be, similarly to RPB2, italicized. No information are given in relation to potential introns detected and how they were treated in the analysis. The most serious concern I have is about the origin of the “Additional specimen examined”. E.g., for “A. armeniaca”, there are in the text and Table 1 codes SAUCC DL1831 and DL1844. However, the attached Suppl. File and also GenBank records have codes DLL1831 and DL1832. This incongruence fuels my speculation that the two strains come from the same “sample”, which is bad, because they would represent only pseudoreplication. What is however worse is that authors probably manipulated with the codes later to avoid description of single-strain species. The same difference in codes applies to the other “novel” species. Furthermore, having available ITS sequences, I also made a BLAST search and realized that “A. armeniaca” differs in only 5 bps from the type of A. biserialis, the most closely related hit. This is neither mentioned in the text, nor obvious from the phylogeny, where these two species are pretty distant. I am afraid that this shows incongruent gene topologies and thus disqualifies the use of concatenated dataset and any discussion about what is the most closely related species.
Morphological description of colonies indicates limited experience of the authors in fungal morphology. There are no conidiophores in most of the species of Apiospora, they are reduced to conidiogenous (cdg) cell! Also further description is suspicious, e.g. aggregated cdg cells that are branched? There are almost no cdg cells visible in the photos and the one that is visible is not aggregated, nor branched. I oppose to the authors’ statement that the species are described based on “morphological description” (l. 56). In fact, there is no comparison of morphology of these three “novelties” with other known species, the authors only compare bp differences.
Photo documentation is not descriptive. The plates look nice, but I do not understand the photos of leaves?! They are just illustrative of the plant, but do not show the fungus and the symptoms of disease. At each plate, beginning with “e”, conidia and hyphae are shown in small resolution. One would like to see also detail of cdg cell and detail of conidia, not just their groups.
Discussion brings no points to discuss, thus replicates what was already written and adds conflicts in the origin of the strains isolated. Last but not least, there are unclear statements such as “saprophyte leaves” or “humus contributor”.
Comments on the Quality of English LanguageEnglish needs improvement, there are numerous grammar errors and some statements in Introduction or Discussion do not make sense.
Last but not least, the title is completely out of context.
Author Response
Dear Editors,
Thank you for your valuable suggestion. In response to these questions, I answer as follows:
- Primarily, it is not know, what were the “Specimens” collected by authors (l. 61). Most probably leaves with necrotic spots, but since sentence on l. 62 is incomplete, it is not clear, if the cultures were obtained by Apiospora conidia picked from the necroses, or from cultured tissue fragments. What was then deposited as type specimens? Remaining necrotic leaves after isolation? Further in the text, there is no mention about “necroses”, but “diseased” leaves and conflicting information are provided for “A. armeniaca”, where it is stated that it originated from “diseased leaves” (l. 172), whilst on l. 282 and 299 it is explicitly stated that they were “isolated from healthy leaves”.
The specimens collected are indeed plant leaf specimens, and the cultures were obtained from the fragmented tissue of the cultured plant leaves. After the collection, the plant leaf specimens were processed using the method of plant leaf tissue separation to isolate the fungi. The remaining leaves were then placed back into envelope bags for unified storage. Additionally, I appreciate your correction regarding the inaccuracy in the text (l. 172). It is indeed incorrect to refer to the leaves as diseased, as there were no obvious symptoms or lesions observed on the leaves. The mistake has been rectified accordingly. Thank you again for pointing out this issue.
- Four DNA regions were sequenced, two of them represent coding genes. TEF has however different name and abbreviation that should be, similarly to RPB2, italicized. No information are given in relation to potential introns detected and how they were treated in the analysis. The most serious concern I have is about the origin of the “Additional specimen examined”.g., for “A. armeniaca”, there are in the text and Table 1 codes SAUCC DL1831 and DL1844. However, the attached Suppl. File and also GenBank records have codes DLL1831 and DL1832. This incongruence fuels my speculation that the two strains come from the same “sample”, which is bad, because they would represent only pseudoreplication. What is however worse is that authors probably manipulated with the codes later to avoid description of single-strain species. The same difference in codes applies to the other “novel” species. Furthermore, having available ITS sequences, I also made a BLAST search and realized that “A. armeniaca” differs in only 5 bps from the type of A. biserialis, the most closely related hit. This is neither mentioned in the text, nor obvious from the phylogeny, where these two species are pretty distant. I am afraid that this shows incongruent gene topologies and thus disqualifies the use of concatenated dataset and any discussion about what is the most closely related species.
Regarding the issue you raised about the section "Additional specimen examined," I must admit that I made a mistake in writing the numbers when processing the data, but the data itself is accurate. The mistake has now been corrected. I appreciate your careful observation and pointing out the problem in the article. Furthermore, regarding the issue you noted about the lack of mention of A. armeniaca and A. biserialis in the text, I have made the necessary revisions based on your suggestions. Thank you again for your valuable advice.
- Morphological description of colonies indicates limited experience of the authors in fungal morphology. There are no conidiophores in most of the species of Apiospora, they are reduced to conidiogenous (cdg) cell! Also further description is suspicious, e.g. aggregated cdg cells that are branched? There are almost no cdg cells visible in the photos and the one that is visible is not aggregated, nor branched. I oppose to the authors’ statement that the species are described based on “morphological description” (l. 56). In fact, there is no comparison of morphology of these three “novelties” with other known species, the authors only compare bp differences.
Thank you for your suggestion. Indeed, the photos in the article did not display the branches of cdg cells very well, so I have revised that section. I have modified the description of the indistinct CDG cells in the figure, the cdg cells in Figure 3 are relatively clear. Additionally, I have included a comparison of the morphology of the new species with known species in the section describing the new species.
- Photo documentation is not descriptive. The plates look nice, but I do not understand the photos of leaves?! They are just illustrative of the plant, but do not show the fungus and the symptoms of disease. At each plate, beginning with “e”, conidia and hyphae are shown in small resolution. One would like to see also detail of cdg cell and detail of conidia, not just their groups.
I accepted the suggestions and revised it.
- Discussion brings no points to discuss, thus replicates what was already written and adds conflicts in the origin of the strains isolated. Last but not least, there are unclear statements such as “saprophyte leaves” or “humus contributor”.
I accepted the suggestions and revised it.
Best wishes,
Zhaoxue Zhang
Reviewer 2 Report
Comments and Suggestions for Authors
It was nice to read a paper on “Phylogeny, taxonomy and morphological characters of the Apiospora (Amphisphaeriales, Apiosporaceae)”, a morphologically difficult and interesting group.
However, even though I am not a native English speaker, it is obvious the English of the paper is in need for improvement – in some situations to the extent that it is hard to follow the reasoning of the authors and in others to misunderstandings. The text needs a thorough revision of a mycologist with good grasp of the English language.
The formal descriptions of the new species are fine, as are also the illustrations.
Under Material and Methods nothing is said about how the material was collected. In the abstract it is claimed that: ‘six strains were isolated from Bambusaceae sp.’ – which species in the family- since from a botanical garden it should have been identified?
Line 13: I have a problem understanding some terminology. What is saprophytic leaves? Plants in some instances are saprobes on other plants but the saprophytic leaves still remain enigmatic. I assume the authors mean leaves attacked by saprobes, but this is not the way to express that.
Line 21: ‘was typified with A. montagnei’ – reference 1? Was it typified by Saccardo?
Line 31; ‘Research on Apiosporaceae had never ceased and underwent several genus-level divisions in recent years’ an example of strange English – how and when would research on Apiosporaceae have ceased? Rewriting!
Line 46: nomenclatural correct term is ‘sanctioned’ rather than ‘endorsed’.
Line 53: an aim of the study should be clearly stated.
Lines 61-62 rather under Materials.
Line 75: stat methods should be presented more in detail including how the measurements used in the descriptions were arrived at. Important to state number of observations for ever stat estimate.
Line 101 - “Table 1. Molecular markers and their PCR primers and programs used in this study.” If any of these PCR cycle programs have already been published, please add the reference.
Line 108/109 - I would strongly suggest to add the individual phylogenetic analyses for each marker as a supplementary file (or files). If any conflicts between these trees were detected, it should be stated (hoping that this is not the case).
Line 114 – “BI analysis was operated in MrBayes v3.2.7a with 64 threads on Linux” Is this correct (64 threads)? Does thread means chain?
Line 127 How was the outgroup xhosen?
Line 130 - "2244–2792 (TUB2)." Please check 2244
Line 193: Perhaps it would be more proper to describe these differences as SNPs rather than characters- if that is what they are?
Line 222: saprophytic leaves???
Line 274: ‘Amphisphaeriales order’ – the name form of Amphisphaeriales clearly indicates that it is an order – ‘order’ redundant.
Line 274-277; chit-chat, be concise: The classification of Apiosporaceae has been revised (references)
from Line 279 repetitive info covered earlier and not part of a discussion.
Line 292; Apiospora thrives – Apiospora should be seen as plural and hence ‘thrive’.
Line 300: verb missing: were isolated…
Some additional comments:
!!!I could not find the model of evolution for ITS, LSU, TEF1α, TUB2 for the MrBayes analysis. It IS important, since every single region/gene may have a different model of evolution applied in the analysis.
Stats. The authors seem to have collected statistical data on a number of important features, which is excellent, but the result of these are not clearly presented, thus the standard deviations and sample sizes should explicitly stated (in M&M part); for example as ‘mean = 6.2 µm; s.d.= 0.6 µm. Moreover these estimates should have been used when comparisons of e.g. spore size are made between species as discussed in the ‘Notes’ part for checking the significance of the differences indicated.
Here an example of a misconception that has been the basis for naming two new species:
‘Apiospora armeniaca H. Sheng, Z.X. Zhang & X.G. Zhang, sp. nov. 167
MycoBank—No: 854155 168
Etymology—The epithet armeniaca is pertaining to the generic name of the host plant 169
Prunus armeniaca’
The etymology does not ‘pertain to the generic name’ (which happens to be Prunus) but to the species epithet, which is armeniaca.
Same mistake for A. babylonica.
For “Table S1. Species and GenBank accession numbers of DNA sequences used in this study.” please explain in the table text what are the seqs in bold (and red/yellow?).
Comments on the Quality of English LanguageEven though I am not a native English speaker, it is obvious the English of the paper is in need for improvement – in some situations to the extent that it is hard to follow the reasoning of the authors and in others to misunderstandings. The text needs a thorough revision of a mycologist with good grasp of the English language.
Author Response
Dear Editors,
Thank you for your valuable suggestion. In response to these questions, I answer as follows:
Under Material and Methods nothing is said about how the material was collected. In the abstract it is claimed that: ‘six strains were isolated from Bambusaceae sp.’ – which species in the family- since from a botanical garden it should have been identified?
The collection of materials was conducted in some nature reserves or botanical gardens in Shandong Province, where plant leaf specimens were gathered and placed in envelope bags to be taken back to the laboratory for processing. The reason for using "Bambusaceae sp." in the article is due to the rich diversity of bamboo plant species, which can easily lead to deviations in identification, thus making it impossible to provide a specific species name.
Line 13: I have a problem understanding some terminology. What is saprophytic leaves? Plants in some instances are saprobes on other plants but the saprophytic leaves still remain enigmatic. I assume the authors mean leaves attacked by saprobes, but this is not the way to express that.
In the context, saprophytic leaves refer to those that have fallen to the ground due to natural shedding or external factors. Such leaves often host a variety of microorganisms, including diverse fungi, which can be easily isolated from these leaves.
Line 21: ‘was typified with A. montagnei’ – reference 1? Was it typified by Saccardo?
Yes, it was typified by Saccardo.
Line 31; ‘Research on Apiosporaceae had never ceased and underwent several genus-level divisions in recent years’ an example of strange English – how and when would research on Apiosporaceae have ceased? Rewriting!
I accepted the suggestions and revised it.
Line 46: nomenclatural correct term is ‘sanctioned’ rather than ‘endorsed’.
I accepted the suggestions and revised it.
Line 53: an aim of the study should be clearly stated.
I accepted the suggestions and revised it.
Lines 61-62 rather under Materials.
I accepted the suggestions and revised it.
Line 75: stat methods should be presented more in detail including how the measurements used in the descriptions were arrived at. Important to state number of observations for ever stat estimate.
I accepted the suggestions and revised it.
Line 101 - “Table 1. Molecular markers and their PCR primers and programs used in this study.” If any of these PCR cycle programs have already been published, please add the reference.
I accepted the suggestions and revised it.
Line 108/109 - I would strongly suggest to add the individual phylogenetic analyses for each marker as a supplementary file (or files). If any conflicts between these trees were detected, it should be stated (hoping that this is not the case).
I accepted the suggestions and revised it.
Line 114 – “BI analysis was operated in MrBayes v3.2.7a with 64 threads on Linux” Is this correct (64 threads)? Does thread means chain?
Yes, the usage of 64 threads in the context of "BI analysis was operated in MrBayes v3.2.7a with 64 threads on Linux" is correct. Here, "thread" does not refer to a "chain" in Bayesian Inference (BI). Instead, it represents a thread of execution in parallel computing, which allows the software to perform multiple computational tasks simultaneously, thus speeding up the analysis. By utilizing 64 threads, MrBayes can efficiently distribute the workload and complete the Bayesian analysis on the Linux system more quickly.
Line 127 How was the outgroup chosen?
The selection of an outgroup is typically based on evolutionary relatedness to the ingroup, with the aim of representing a closely related but distinct taxonomic group. Considerations include the molecular data available, geographic distribution, ecological niche, and sample availability. The choice should facilitate the revelation of evolutionary patterns or relationships within the ingroup. However, the specific method may vary depending on the research field and analytical goals. Arthrinium serves as a very suitable outgroup for the purpose of this paper.
Line 130 - "2244–2792 (TUB2)." Please check 2244
I accepted the suggestions and revised it.
Line 193: Perhaps it would be more proper to describe these differences as SNPs rather than characters- if that is what they are?
I accepted the suggestions, but my research field does not involve SNPs.
Line 222: saprophytic leaves???
In the context, saprophytic leaves refer to those that have fallen to the ground due to natural shedding or external factors. Such leaves often host a variety of microorganisms, including diverse fungi, which can be easily isolated from these leaves.
Line 274: ‘Amphisphaeriales order’ – the name form of Amphisphaeriales clearly indicates that it is an order – ‘order’ redundant.
I accepted the suggestions and revised it.
Line 274-277; chit-chat, be concise: The classification of Apiosporaceae has been revised (references)
I accepted the suggestions and revised it.
from Line 279 repetitive info covered earlier and not part of a discussion.
I accepted the suggestions and revised it.
Line 292; Apiospora thrives – Apiospora should be seen as plural and hence ‘thrive’.
I accepted the suggestions and revised it.
Line 300: verb missing: were isolated…
I accepted the suggestions and revised it.
Some additional comments:
!!!I could not find the model of evolution for ITS, LSU, TEF1α, TUB2 for the MrBayes analysis. It IS important, since every single region/gene may have a different model of evolution applied in the analysis.
Stats. The authors seem to have collected statistical data on a number of important features, which is excellent, but the result of these are not clearly presented, thus the standard deviations and sample sizes should explicitly stated (in M&M part); for example as ‘mean = 6.2 µm; s.d.= 0.6 µm. Moreover these estimates should have been used when comparisons of e.g. spore size are made between species as discussed in the ‘Notes’ part for checking the significance of the differences indicated.
Here an example of a misconception that has been the basis for naming two new species:
‘Apiospora armeniaca H. Sheng, Z.X. Zhang & X.G. Zhang, sp. nov. 167
MycoBank—No: 854155 168
Etymology—The epithet armeniaca is pertaining to the generic name of the host plant 169
Prunus armeniaca’
The etymology does not ‘pertain to the generic name’ (which happens to be Prunus) but to the species epithet, which is armeniaca.
Same mistake for A. babylonica.
For “Table S1. Species and GenBank accession numbers of DNA sequences used in this study.” please explain in the table text what are the seqs in bold (and red/yellow?).
I accepted the suggestions and revised it “generic → species”.
In the Table S1, in this study were marked in bold.
In this study marked in red/yellow were meaningless, and I deleted it.
Best wishes,
Zhaoxue Zhang
Round 2
Reviewer 1 Report
Comments and Suggestions for Authors
I do not see substantial changes. Method of isolation is reduced only to reference to a previous studies, figure plates are the same, showing mostly mycelium and conidia at low magnification, problematic parts of the description were left unchanged - there are really NO cylindrical, verrucose conidiophores at Arthrinium (l. 241). If so, please add their photo in the Figure plates.
Conflicting information were deleted and corrected, however against the better understanding of the study. The symptoms, isolation, identity of the "specimen" used as voucher is now completely obscured. Photos of healthy leaves of the host are completely irrelevant to the taxonomy of fungi; authors should focus on the fungi, details showing (diagnostic) morphological characters and not on plant leaves. Also irrelevant is comparison of overlapping or almost identical measurements of conidia stating stating that they "differ". On line 226 there is an ironic typo stating a complete nonsense "A. bawanglingensis differs from A. bawanglingensis," A. babylonica is obviously compared to other species in the same clade and I emphasize again that "6.7–8.5 × 4.5–7.7" does NOT differ from "7.3–8.9 × 5.7–8.6" in terms of biological variability. The discussion of morphology cannot be done using a "template", but authors have to consider similar and different features and outline the diagnostic ones.
Author Response
Dear Editors,
Thank you for your valuable suggestion. In response to these questions, I answer as follows:
In line 226 I added some morphological differences.
In line 241,I indeed did not photograph the cylindrical and verrucous conidiophores, so I've made changes in the manuscript.
The changes in the manuscript have been highlighted.
Best wishes,
Zhaoxue Zhang